# Small-Town Citizens' Technology Acceptance of Smart and Sustainable City Development

**Giovanni Baldi \***, **Antonietta Megaro and Luca Carrubbo**

Department of Management & Innovation Systems, University of Salerno, 84084 Fisciano, Italy
\* Correspondence: gbaldi@unisa.it

**Abstract:** Citizens are an essential part of the process of smartification and sustainable development of cities as they must adopt, understand and interact with the enabling technologies of digital transformation of societies, cities, and public administration. Therefore, technology acceptance is crucial to creating smart and citizen-centered cities. This is even more challenging in small towns that suffer from an aging population, desertification, lack of infrastructure, and especially the digital divide. The purpose of this research is to investigate the adoption of an Urban Services Technology (UST) in tourism management within a tourism-oriented small town in Southern Italy. A questionnaire was then constructed according to the 12-variable Urban Service Technology Acceptance Model (USTAM), and 216 responses were obtained from a defined group of 1076 subjects. Analyzing the data with a quantitative approach by conducting Exploratory Factor Analysis (EFA), Confirmatory Factor Analysis (CFA), and Structural Equation Modeling (SEM), the assumptions of the initial model were all rejected and new five factors emerged. The path diagram shows that only the factors Sustainability, Ease and Value have a positive correlation with technology adoption. Future research might investigate the mediating role of socio-demographic variables on technology acceptance by considering geographical and cultural diversity among small towns.

**Keywords:** smart city; small town; smart citizens; citizen engagement; technology acceptance

## 1. Introduction

The digital revolution is significantly changing daily routines, interpersonal relationships, and quality of life in medium-sized to large cities, where every activity, including innovation, sustainability, and resilience, will take place [1]. As well, it is realistic to assume that in the next decades, a sizable chunk of the digital economy and beyond will be played in these comparable sites [2,3]. Since the development of ultrabroadband networks and the creation of new services in urban areas were seen as being closely related, it was no accident that the issue was included in the European digital agenda [4,5]. The emergence of 5G, the Internet of Things, AI, and smart grids are strongly pushing in that direction, enabling an increasing number of new technologies and services developed and produced by thousands of start-ups that are essentially building a parallel economy to the traditional one, dense with the future and that, not surprisingly, tends to condense around large urban hubs, tying inseparably with the research activity carried out by universities and big businesses [6,7]. In this regard, it does not seem out of place for those who have long advocated for a new urbanism as a catalyst for growth and a new humanism, in which new services put Man back at the center [8].

According to some scholars, the concept of a smart city is holistic in character; that seems to be, it begins with the creation of residential and commercial structures as elements of a broader whole that may create outcomes greater than their total when taken separately [9,10]. As it turns out, a smart city is created by the combination of multi-layered smart buildings and smart citizens rather than the other way around [11–13]. Exactly, for

this reason, it is claimed that even small communities could see smart growth that begins with their citizens' engagement [14–17].

Smaller, dispersed metropolitan centers appear to benefit from technology, but in general rural regions frequently lack key infrastructure such as roads, running water, power, and high-speed cable Internet [18]. It should come as no surprise that when dealing with networks, the approach must eschew the radicalism of individual technologies and instead seek the best solution on a case-by-case basis [19]. In fact, a city's digital infrastructure is more comparable to a mille-feuille cake than a solid foundation, allowing each person to select the connectivity that will best serve their needs [20,21]. As a result, there will be a wide range of participants, as new apps increasingly stem from the actions of private individuals rather than only official bodies as the major actors [22,23]. Information-based deregulation is increasingly being shown to be far more successful than top-down management in managing the spaces of life between home and work. In other words, the provision of information in real-time already enables citizens to self-regulate, to avoid potentially dangerous situations [24], or to change their behavior; but also, to save money, for instance, by forgoing ownership of assets when their typical use is extremely limited; the sharing of electric vehicles is a striking example of this [25]. In other words, tackling the share economy merely requires acquiring "smart citizenship." Smart citizenship enables the citizen to change from a passive subject to an agent in the community where they live and work [26]. In order for us to really talk about a smart city, it is essential to put the city user back at the center, directly involving him or her in this new way of understanding and building the common good [27]. In fact, technology is a means to simplify people's lives and not the end. The human factor and the technological factor cannot and should not be separated [28,29]. Then, we no longer speak only of Smart City but of Smart Citizenship, as a dimension where people are the bearers of a citizenship, intelligent and proactive, which is realized through new forms of participation in which the implications of the digital revolution are shown clearly [26,30–33].

Beyond its scale, smart city implements civic living consciousness by leveraging increasingly fine-grained and precise knowledge obtained through the statistical analysis of enormous data created by the people and things that determine the city itself [34]. Technology is at the foundation, but people and their demands for social life are at the core [35]. Social life as a manifestation of social constructivism. According to this theory, the social construction of reality is an ongoing, dynamic process, and people's actions are how reality is reproduced [36]. To summarize, it is widely acknowledged that all of this is also causing concern, driven by the unease of a segment of the public who feels unable to keep up with a technological advancement that is not always "friendly," and that the idea that "machines" operate independently of humans is both intriguing and unsettling [37]. In this context, policymakers and the general public must play an unassignable role by establishing informative and mediating mechanisms with the express goal of ensuring technical inclusiveness as a precursor to inclusivity tout court [38]. However, even people who are oblivious to them may clearly perceive the advantages.

The smart city concept is not limited to major municipalities; it can also be used in smaller towns [9,39]. There is therefore a need for the efficient use of technology to provide services to citizens, some of whom may have direct contact with the local government [40]. However, the building of a smart city is sometimes hard, and the cost is frequently prohibitive or excessively expensive [41]. However, it would suffice to begin with the use of data that is now available but frequently underutilized or completely neglected in order to create the conditions for small- to medium-sized urban realities to become smart [42,43]. In this regard, a number of businesses are emerging that gather information from various municipalities and provide services like map hosting for monitoring safety warning zones and limited traffic zones, as well as digital services for residents, tourism industry professionals, and tourists [44–46].

According to Štefkovičová and Koch [47], it is crucial to establish these settings for these residents since even small towns and rural populations desire to be smart. There

are several outstanding case studies, such as the management of cultural resources in a historic Norwegian town [48] or where intelligent, sustainable heating and lighting systems have been created [49,50]. According to other studies, it is imperative to involve important stakeholders in the process of transformation from a small to a smart and sustainable city [12,22,51–54]. The focus of smart city development in the Finnish small cities researched by Ruohomaa et al. [51] was observed to be shared e-bikes (smart mobility), waste management services (smart environment), and robots in education and elderly care (smart living). Contrarily, it is far more difficult to structurally integrate small towns and rural areas into a smart city development network in exceptionally large countries such as Canada [55]. Finally, Zavratnik et al. [56] emphasized the significance of communities' essential position in development processes, hastened the need to comprehend communities within particular contexts, and illustrated how sustainability for the future can only be achieved by active citizen participation. We can attain sustainability for the future by actively engaging the community that is, people [56,57].

If the individuals who are supposed to benefit from smart city infrastructure and technology do not use them or do not know how to use them, they are rendered ineffective [58]. Furthermore, successful technology adoption is required for the development of citizen-centric smart cities [59].

Based on these considerations, the purpose of this study is to assess the acceptability of urban service technology (UST) applied to tourism in a small town.

The study is organized as follows: the first section is a review of the literature on technological acceptance models used in the Smart City setting, as well as versions that integrate social variables. Then, in part 3, we discuss the study's context and methods. Section 4 displays the research findings as well as the path diagram. Finally, in the final section, we have added the study's discussions, implications, and limitations.

## 2. Literature Review

### 2.1. Technology Acceptance Model Applied to Smart Cities

Since place-based aspects of technology adoption are crucial to understanding how technology may be successfully adapted and applied to more varied human populations, the Technology Acceptance Model has previously been employed when addressing technologies for smart cities [60].

Researchers especially employ the Technology Acceptance Model (TAM), which Davis [61] created based on Fishbein and Ajzen's theory of reasoned behaviors [62], to better understand how people perceive and adopt recent technology. The factors examined in this model—which is also the most popular in smart cities and technology—are: Perceived Security (PS), Relative Advantages (RA), Perceived Ease of Use (PEU), Perceived Usefulness (PU), Compatibility (Co), Reliability (Re) [61].

Davis [61] defined Perceived Usefulness as "the degree to which a person believes that use of a particular system would enhance his or her job performance" as opposed to Perceived Ease of Use, which he defined as "the degree to which a person believes that use of a particular system would be free of effort" [63]. The adoption or usage of technology in a TAM is governed by Behavioral Intention (BI). Attitude toward usage, as well as the direct and indirect consequences of PEU and PU, all have an impact on Behavioral Intention. PU and PEU are identified as key determinants of citizen adoption of smart city technology in the majority of studies on this topic utilizing the TAM model [64].

Technologies for smart cities have been studied in relation to smart homes [65], healthcare [66], electronic payment systems [67], smart mobility [68,69], mobile applications [70,71], 5G connectivity [72], to e-governance platforms [73], heritage education and management [74,75], and smart technology for cities generally [76]. There was no research conducted on whether any technology for SCs would be accepted in the tourism industry.

Indonesia and India were the nations where the most studies were conducted [64]. The few studies that have been conducted in Europe have been in Oslo and Tallinn [77], a case study combining Berlin, Dublin, London, Milan, and Madrid [76], and Luxembourg [75].

We are unable to locate any research on TAM applied to the smart city that was conducted only in Italy, much less in rural or small towns.

### 2.2. Social Factors Influencing the Acceptance of Technologies for Smart Cities

TAM has been debated, employed in plenty of academic research in different fields, and has undergone numerous revisions and implementations along with technological advancement. In actuality, more variations are accessible since more factors have been introduced throughout time. We discover the TAM2 model of Venkatesh and Davis [78] by including the variables of experience and voluntariness that modify the influence of subjective norms on planned usage. In order to forecast individual usage of information technology, the TAM3 model was developed in 2008 by Venkatesh and Bala [79], going further into the concept of perceived ease of use. In reality, it includes other factors including computer playability, anxiety, and self-efficacy.

The Unified Theory Acceptance and Use of Technology (UTAUT), which accurately describes the acceptance of technology for information systems, was afterward created from the latter model. The establishment of Behavioral Intention is influenced by a number of factors, including Performance Expectations, Effort Expectations, Social Influence, and Enabling Circumstances. These are adversely associated with other factors including Voluntariness of Usage, Gender, Age, and Experience, according to research [80].

According to Prasetyo and Santiago [81], the Enabling Circumstance has a significant impact on the Behavioral Intention (BI) of individuals who work in smart cities all over the world. This analysis suggests that respondents' BI to continue working, show up to their jobs every day, and to perform successfully and efficiently is highly impacted by how the conveniences of the work and living environment may fit them.

TAM fully disregards product attributes and social aspects in favor of emphasizing the user's subjective attitude and usage behavior [82]. As a result, other TAM extensions emphasize the significance of Social Influence (SI). TAM has been significantly expanded and employed to forecast driver adoption of technology and driving support systems, claim Zhang et al. [83]. In addition, Wang et al. [84] argue that TAM has been adopted to study consumer reactions to Autonomous Vehicles (AVs). When Park et al. [69] looked into Social Influence specifically, they discovered how exactly this characteristic, together with easy conditions and perceived utility, are crucial in people's intentions to use autonomous cars in SCs, and that demographic factors might instead lessen such intentions. To examine the level of adoption and acceptance of SCs technologies, however, all studies incorporate new variables from the aforementioned models, such as User Experience, Internet quality, attitude, sociodemographic factors, culture, quality services, awareness, trust, and security [64].

### 2.3. Social Cognitive Theory (SCT) and the USTAM model

SC's epistemic foundations, rooted in command-and-control theory and scientific management, lead to a very traditional and technocratic view of urban management and governance [85]. However, new urban challenges cannot be addressed only by methods of greater efficiency. These challenges also—and probably—concern sustainability and resilience and require new and innovative approaches to urban governance [86]. Such approaches will have to involve the 'human factor,' cognition, creativity, and the ability to learn to cope with disruptive changes using technology that has cognitive capabilities that people often connect with the human mind [87]. Moreover, cities are complex socio-technical systems [88], so their challenges cannot be addressed through technological developments and innovations alone. In fact, Finger and Portmann as early as 2016 introduced the concept of cognitive cities [89].

Service intelligence that is focused on individuals and citizens also contributes to the notion of the cognitive city [60]. The deployment of these might alter the state of data intelligence today to satisfy societal needs for urban services [90]. The capacity to develop smart cities from the perspective of infrastructure, human dynamics, human

comprehension and prediction, and human-machine interfaces to social sustainability might be improved with a people-centric perspective [91].

The given information thus far is based on Social Cognitive Theory (SCT). This theory focuses more on the social and economic aspects of technological acceptance. The most popular hypothesis to describe socially conscious technology is the social cognitive theory [92]. The social cognitive theory's ideas may be used to better understand how people behave when it comes to accepting new technologies. It focuses mostly on examining how society affects a person's conduct. It is one of the most significant ideas for elucidating human behavior, according to Bandura [92]. Grasp the social aspects of technological acceptance also requires an understanding of innovation diffusion theory.

TAM and SCT were developed and integrated through information and communication systems research, making them reliable theories for gauging people's opinions of using new technology [93].

As a result, a new acceptance model for Urban Services Technologies (UST) called Urban Services Technology Acceptance Model (USTAM) was theorized and confirmed [94] precisely because citizens can be influenced by social factors such as Work Facilitating, Cost Reduction, Energy Saving and Time Saving [95].

The USTAM model for SCs is composed of the following factors as seen in Figure 1:

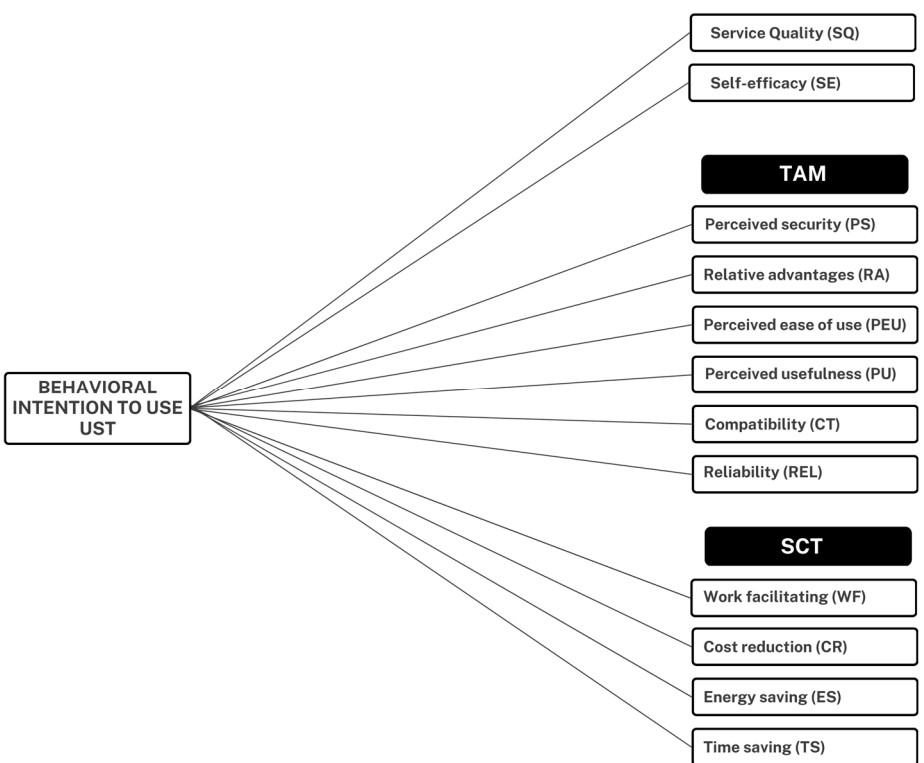

**Figure 1.** USTAM model [94].

### Perceived Security (PS)

According to research, people in developing nations have a great need to feel secure when utilizing modern technologies [94]. The degree to which consumers consider smart city technologies or services to be a safe platform for storing and sharing personal data is known as perceived security [96]. The adoption of innovations is hindered by a lack of perceived security [97] and external pressure and perceived information security influence trust in smart city technologies [98]. Previous research on this topic has been conducted in several fields such as cloud computing, education, e-banking, and e-governance services [99–102]. Users of smart city technologies are likely to prioritize safety and security, according to research [103]. As a result, this study would like to suggest the following:

**Hypothesis 1 (H1):** *PS positively influences UST users' intention.*

### Relative Advantage (RA)

Comparing the new technology to what is now available uses the concept of relative advantage [94]. According to studies, the relative advantage is the most accurate indicator of technological adoption [104,105] and enables better functioning of the city and city life [106] as a user must believe that one technology is better than those before in use [107]. As a result, this paper would like to propose the following idea:

**Hypothesis 2 (H2):** *RA has a favorable impact on UST users' intention.*

### Perceived Ease of Use (PEU)

The degree to which people think utilizing smart technology involves a lot of effort is known as perceived ease of use. According to various research [108,109], perceived ease of use is a key component in determining how well-liked government e-portals and other technologies are such as digital payments [110]. The following hypothesis is what this research would want to suggest going forward:

**Hypothesis 3 (H3):** *PEU influences UST user's intentions.*

### Perceived Usefulness (PU)

According to the concept of perceived utility, individuals may be more likely to adopt smart city technology if they believe it would simplify their lives [111]. Government and major organization bureaucracy can at times be highly intricate, very time-consuming, and follow-up in emerging nations [94]. In light of it, this paper would like to provide the following idea:

**Hypothesis 4 (H4):** *PU influences UST user's intention.*

### Compatibility (CT)

The perception of technology compatibility, which significantly influences utilization, might represent the idea of creating smart cities [111]. The level of fit between a smart urban technology and potential consumers' current behavior is known as compatibility [94]. The interoperability of the new technology with the consumers' existing gear or software is even more crucial [112]. Research has also shown that technical compatibility is the key to external diffusion of the technology, whereas relative advantage would be the key to internal diffusion, as found by Van Oorschot, et al. [113]. They found that technical compatibility is a strong predictor of technology acceptance. As a result, this study would like to suggest the following:

**Hypothesis 5 (H5):** *UST user's intention is benefited by CT.*

### Reliability (REL)

Users' trust in the functionality and correctness of the technical service is explained by its reliability [94,96]. For a user to consistently utilize a technology, they need to believe that it is dependable [103]. In light of the fact that customers value reliability, the following statement is what this paper would want to make:

**Hypothesis 6 (H6):** *UST user's intention is positively impacted by REL.*

### Self-efficacy (SE)

Self-efficacy illustrates how well a person can use technology to conduct specific activities. It speaks to a person's assurance that they can use a technology efficiently [114,115]. It is suggested that those who have a high level of technological self-efficacy can utilize digital

technologies more regularly and do so with less anxiety. In light of this, the following theory is put forth:

**Hypothesis 7 (H7):** *UST user's intention is positively impacted by SE.*

### Service quality (SQ)

Tangibility, Recovery, Responsiveness, and Knowledge were identified to be the key SQ construct aspects in the service factory [116]. The decision of an urban citizen to consistently utilize a technology for SCs might be influenced by Service Quality [94]. In fact, the inclination to utilize technology really grows the more highly the service is seen to be [96]. This leads to a new research hypothesis that is as follows:

**Hypothesis 8 (H8):** *SQ impact positively UST user's intention.*

### Work Facilitating (WF)

Users' perceptions of the technical infrastructure's suitability to support them while utilizing technology are referred to as facilitating conditions [94]. According to research, a successful adoption of technology is favored by good working circumstances and the belief that using UST will make completing daily chores more efficient [117]. Consequently, we would like to provide the following hypothesis in this paper:

**Hypothesis 9 (H9):** *WF influences UST user's intention.*

### Cost Reduction (CR)

Due to users' perceptions that they can reduce present expenses, cost reduction indicates that the new technology will result in new economic benefits [118,119]. Van Oorschot, et al. [113] claim that the study demonstrates the importance of cost in numerous technological studies. As a result, the study would like to suggest the following:

**Hypothesis 10 (H10):** *CR has a favorable impact on UST user's intention.*

### Energy Saving (ES)

Energy conservation refers to the users' feeling that they can save energy by using any smart city technology [63]. To avoid energy loss, this would be a key construct to relate [102,118]. Therefore, this paper would like to formulate the following hypothesis:

**Hypothesis 11 (H11):** *ES has a positive effect on UST users' intention.*

### Time Saving (TS)

By utilizing any smart city technology, people feel as though they are saving time and becoming more time efficient. Conditions that reduce time wastage can affect consumers' adoption of the new technology, according to research by Chiu et al. [118] As a result, the following theory is put forth:

**Hypothesis 12 (H12):** *UST user's intention is positively impacted by TS.*

## 3. Methodology

### 3.1. Context of the Study

The study focuses on Castellabate: a small Italian town and its citizens who are also managers or owners of tourist accommodations from a multi-actor perspective [120]. They use the UST provided by the municipality and owned by a third party (PayTourist) to conduct multiple tasks, to register guests, and to collect tourist taxes. As a result, the City may reap several benefits in terms of monitoring visitor movements and providing facilities

for data collecting, therefore conserving resources and becoming more sustainable and smarter.

The municipality of Castellabate is a town in southern Italy in the province of Salerno and it could best represent the problem of tourism technology acceptance from a small-town perspective towards smartness and sustainability. The questionnaire was sent to all citizens who own, manage or employ a tourist accommodation facility and use the UST. It was possible to send the questionnaire via email to all those who have a facility regularly registered to the platform (1071) thanks to the collaboration with the tourist office and the patentees of the UST. Castellabate has approximately 8000 residents, half of which are active in tourist receiving activities and another significant amount in supporting them. A total of 216 citizens replied with a 20% response rate of the entire target population, and for the reasons stated above, they can be considered a representative sample, as shown in Table 1.

**Table 1.** Sample description.

| Demographical Characteristics | | Frequency | Percentage |
|---|---|---|---|
| Age | 20–30 | 2 | 1% |
| | 31–40 | 43 | 20% |
| | 41–50 | 63 | 29% |
| | 51+ | 108 | 50% |
| Gender | Male | 102 | 47% |
| | Female | 114 | 53% |
| Education | Junior school degree | 35 | 16% |
| | High school degree | 145 | 67% |
| | Bachelor | 26 | 12% |
| | Master | 11 | 5% |
| Type of property | Apartments | 162 | 75% |
| | B&B | 24 | 11% |
| | Guest House | 15 | 7% |
| | Hotel | 15 | 7% |
| Job title | Owner | 168 | 78% |
| | Manager | 17 | 8% |
| | Employee | 17 | 8% |
| | Other | 13 | 6% |
| Who uses UST | On my own | 188 | 87% |
| | Employees | 9 | 4% |
| | I get help from relatives or friends | 11 | 5% |
| | I get help from professionals | 6 | 3% |
| | I get help from tourist office | 2 | 1% |

*3.2. The Questionnaire*

A structured questionnaire was the only instrument utilized in this study's quantitative research methodology to gather data. The questionnaire made on Microsoft Forms includes 6 questions to describe the sample and 43 questions to evaluate the suggested structures and covered the 12-USTAM model constructs depicted in Figure 1 [94].

The questions were rated on a five-point Likert scale. The Likert scale was labeled as follows:

1 = Totally Disagree, 3 = Neutral, 5 = Totally Agree

The Likert scale is a psychometric technique for measuring attitude invented by psychologist Rensis Likert, it is presented, for each item, as an agreement/disagreement scale, with 5 or 7 modes. Respondents are asked to indicate on the items their degree of agreement or disagreement with what the statement expresses [121]. Using this technique, the researcher is able to evaluate the multi-construct data and successfully classify the constructions based on factor loadings [122].

### 3.3. Material and Methods

Statistical analysis was performed using the software RStudio Version 1.4.1717 (The R Foundation for Statistical Computing). The following computations were performed with that software:

(A)  Control of data
(B)  Transformations of variables
(C)  Sampling
(D)  Exploratory Factor Analysis (EFA)
(E)  Cronbach's alpha
(F)  Confirmatory Factor Analysis (CFA)
(G)  Second sampling
(H)  Creation of matrices of variance and covariance
(I)  Export of these matrices

The analysis with a Structural Equation Model (SEM) was continued on the statistical software Lisrel ("LISREL—Scientific Software International, Inc.," n.d.). The dataset had no missing data. Some changes were made to the dataset, one major change being to reverse the negatively placed scales.

The scales of the negatively posed questions were reversed by the following method:
Inverted scale = $Max(L) - x_i + 1$
$Max(L)$ = Maximum on the Likert scale, in our case 5
$x_i$ = value of the response, i.e., the number chosen by the respondent on the Likert scale. The only negatively placed variable is BI3.

Exploratory Factor Analysis (EFA) was conducted to allow us to reduce the set of observed variables to a smaller set of latent variables (factors). No a priori assumptions are made about which factors affect the observed variables; this method, therefore, allows the observed variables to be transformed into a simpler structure that nevertheless contains the same information as the original [123].

Confirmatory factor analysis (CFA) was then conducted to validate the hypotheses made about the relationships between the observed and latent variables; it is then used when one has fairly clear ideas about which factors influence the variables [124]. The estimation method used by default by Lisrel software is maximum likelihood; assuming that the observed variables distribute normally, Lisrel estimates the unknown parameters. The result was then subjected to rotation by various methods. The orthogonal rotation method used here was Varimax because it preserves factor independence [125].

Communality was calculated to describe how much of the variance of the observed variable is explained by the variance of the factor saturating that variable [126]. For the purpose of interpreting the result, it is important to assess the amount of variability or, more precisely, the variance "explained" by the set of factors considered and by each factor individually.

Cronbach's Alpha was then calculated to assess reliability as internal consistency. The closer the Alpha index is to 1, the higher the reliability. Conventionally, an Alpha value above 0.70 is considered acceptable [127].

Through chi-square then we evaluated the null hypothesis of a correct model specification against the alternative hypothesis of an unconstrained matrix of variances and covariances. This approximation is valid under certain conditions: normality of variables, covariance matrix analysis, and sufficient sample size [128].

Finally, we constructed the path diagram to better visualize the correlations between factor X and the independent variable Y (Figure 2).

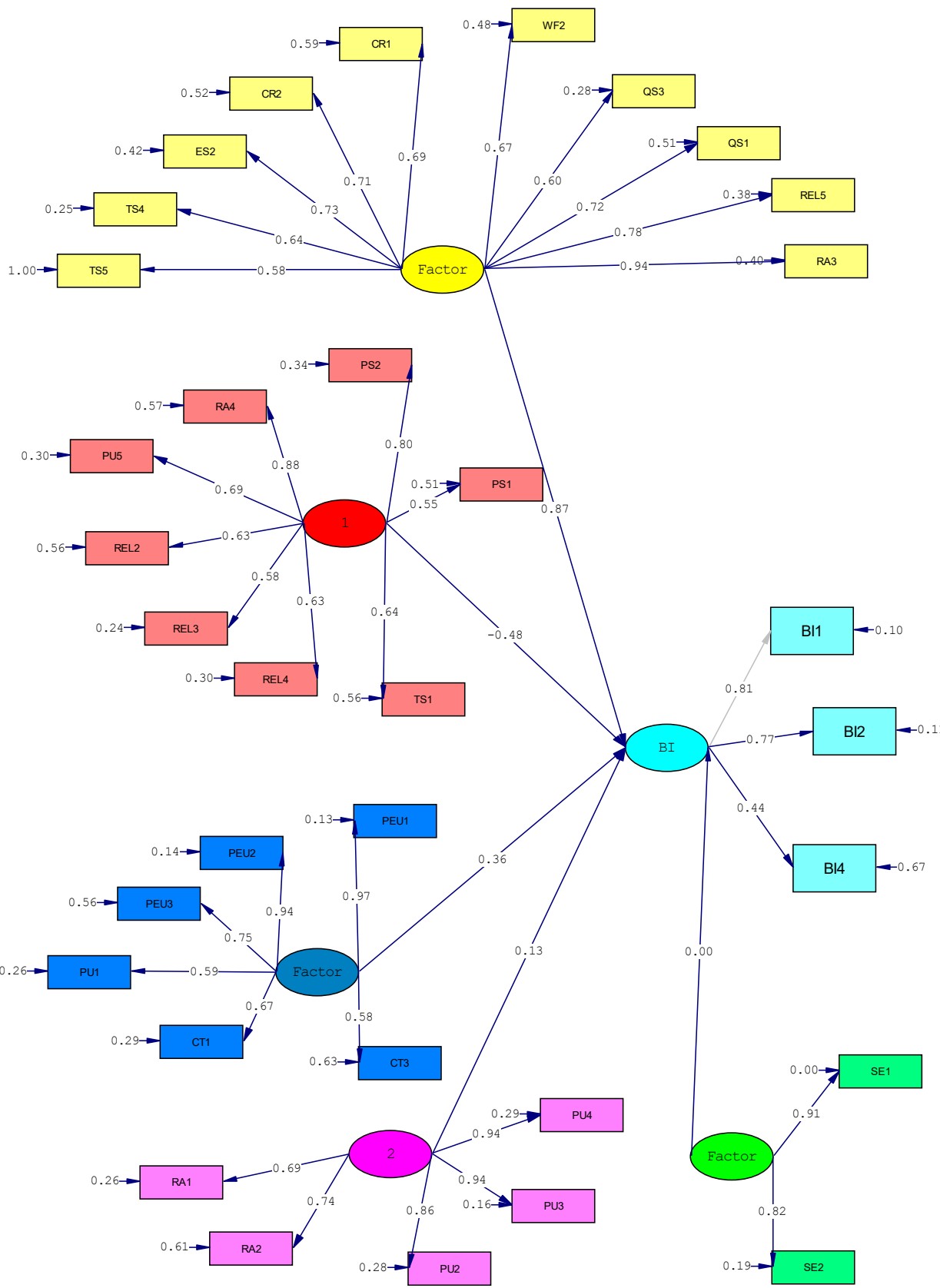

Chi-Square=2296.72, df=512, P-value=0.00000, RMSEA=0.127

**Figure 2.** The path diagram of emerging factors.

## 4. Results

Using the methodology described above, we discovered that the variables in the initial 12-USTAM model did not combine the components, and there were questions distributed across many factors.

Therefore, we can say that the assumptions of the initial model have all been rejected.

Because of this, the program simply grouped the following five components together to find correlations, which may be found in Table 2 with linked questions and Λ (Loading Factors), and Squared Multiple Correlations for Structural Equations of BI is 0.71.

**Table 2.** Emerging construct's loading factor.

| Construct | Item | Measure | Loading Factor |
|---|---|---|---|
| Factor 1 | RA3 | Using UST raises the reputation of my property | 0.94 |
| | REL5 | UST makes me more trustworthy in the eyes of my customers | 0.78 |
| | QS1 | Using UST, I can avoid delays caused by bureaucratic procedures | 0.72 |
| | QS3 | Using UST prevents human errors that can happen with paper-based support | 0.60 |
| | WF2 | It is easy to manage information and communicate using UST | 0.67 |
| | CR1 | I believe UST reduces the costs associated with paperwork | 0.69 |
| | CR2 | I believe UST reduces the cost of activities | 0.71 |
| | ES2 | UST reduces building energy costs as services are delivered remotely, without offices | 0.73 |
| | TS4 | UST allows me to check in (arrival and guest registration) in less time than before using it | 0.64 |
| | TS5 | UST saves me time spent in long lines at administrative offices, post offices, bank | 0.58 |
| Factor 2 | PS1 | UST allows me to complete transactions without harassment | 0.55 |
| | PS2 | I believe that what I do with UST is protected and safe | 0.80 |
| | RA4 | I believe that UST covers what municipalities and citizens need | 0.88 |
| | PU5 | UST is an effective way to interact with my municipality | 0.69 |
| | REL2 | I believe that UST presents accurate and up-to-date information | 0.63 |
| | REL3 | I believe UST is more dependable than physical government offices | 0.58 |
| | REL4 | I believe that UST is dependable | 0.63 |
| | TS1 | I believe that by using UST I can do my tasks faster | 0.64 |
| Factor 3 | PEU1 | I learned to use UST with great ease | 0.97 |
| | PEU2 | I became proficient in using UST | 0.94 |
| | PEU3 | UST platform is easy to use (user-friendly) | 0.75 |
| | PU1 | I am fully familiar with all the features of UST | 0.59 |
| | CT1 | I can use UST from any city, remotely | 0.67 |
| | CT3 | Using UST, I perform my role and tasks faster than interacting with the tourist office | 0.58 |
| Factor 4 | RA1 | The use of UST is necessary for my work | 0.69 |
| | RA2 | I think using UST is very much in line with the way I want to collect information | 0.74 |
| | PU2 | I find the UST platform useful | 0.86 |
| | PU3 | I think UST can offer me a valuable service | 0.94 |
| | PU4 | UST gives me more control | 0.94 |
| Factor 5 | SE1 | I think about using UST features efficiently | 0.91 |
| | SE2 | I think I use UST successfully | 0.82 |

We previously deleted three items from the EFA (WF3, TS2, TS3) due to cross-loading or inadequate load factor, and we then removed four items to increase Cronbach's alpha (ES1, CT2, CT4, BI3).

The communalities value is calculated by using the AVE shared values of all factors presented in Table 2, and the results have been included in Table 3 along with Cronbach's alpha and CR for all factors that emerged from the data analysis.

**Table 3.** Cronbach's alpha, AVE, and CR of the emerging factors.

| Construct | Cronbach's Alpha | AVE | CR |
|:---:|:---:|:---:|:---:|
| Factor 1 | 0.9354444 | 0.570 | 0.812 |
| Factor 2 | 0.9065434 | 0.566 | 0.771 |
| Factor 3 | 0.9234804 | 0.699 | 0.818 |
| Factor 4 | 0.9298417 | 0.734 | 0.820 |
| Factor 5 | 0.9367044 | 0.886 | 0.838 |

Using LISREL software, we constructed the path diagram to better visualize the loading factors and correlations between the latent variables and the variable Y (BI) as can be seen in Figure 2 [129].

## 5. Conclusions

### 5.1. Discussion

The purpose of this research was to develop a model to analyze the acceptance of technologies in small towns that want to become smart and sustainable, considering social, cultural, infrastructural, and demographic factors. The study fills a notable gap in the literature by being unique in that it focuses first on technology applied to tourism, especially in the hospitality industry, and it concentrates on a small town in a rural area of southern Italy with a strong tourist concentration. To do this, we used the USTAM model [94] that had been used for a similar study and seemed to be more appropriate. However, we saw how all the research hypotheses were refuted and how careful statistical analysis could have grouped the items into five new factors.

From what emerged from this analysis and investigating the questions that were asked (see Table 2) we can find a common 'emotion' across the measurements to try to identify emerging factors. The words that could best represent these are:

- Factor 1 = Sustainability
- Factor 2 = Benefits
- Factor 3 = Ease
- Factor 4 = Value
- Factor 5 = Self-efficacy

These factors are remarkably similar to those in the UTAUT model studied in the literature, in which we find variables related to Performance Expectations, Effort Expectations, Social Influence, and Enabling Circumstances [80].

The research using Structural Equation Modeling and the path diagram (Figure 2) reveals that the factors Sustainability (Factor 1), Ease (Factor 3), and Value (Factor 4) positively affect the usage and adoption of tourist technology for small towns seeking to become smart and sustainable.

According to the literature, technology adoption affects the idea of sustainability, which is defined as economic, social, and environmental sustainability, as well as ease of use and perceived value [108–110,130].

On the other hand, Factor 2 (Benefits) does not have a positive relationship with intention to use since its sample of respondents is comprised of older individuals with poor educational levels who do not completely appreciate the possibilities of technology and consistently perceive it as slowing down rather than speeding up activities [131].

For the same reasons mentioned above and in regard to the fact that they are unaware of the full potential of such technology, but they are aware that it might be used better, factor 5 (Self-efficacy) demonstrates an indifferent relationship to intention to use [132].

### 5.2. Implications and Limitations

The implications for future research appear intriguing because it deals with an unexplored topic: small rural towns that want to become smart but do so through the implementation of technologies, particularly those adopted by citizens, who must perceive them

as sustainable, beneficial, easy, effective, and valuable. Rural residents are of old age, have a poor level of education, and are unfamiliar with contemporary technology. Young people in these cities frequently travel to larger cities for education and job, leaving economic and city activities to their relatives, promoting desertification and confounding these processes of sustainable and smart growth.

The current study certainly suffers from the drawback of being confined to one town, focusing on just one technology used in tourism and hospitality, and failing to explain the potential effect of socio-demographic aspects.

Based on these constraints, future research might replicate this research in other cities and see if the citizens behave in the same way, in order to develop a reliable model for small cities. Moreover, future research should look at the mediating role of socio-demographic variables on technology acceptance by considering geographical and cultural diversity among small cities by using the same research model presented here. Scholars can also research the adoption of technology in other smart city areas other than tourism [133].

Finally, it may be useful to conduct qualitative interviews with all of the actors involved in a small urban ecosystem [52], such as the mayor, administrators, policymakers, municipal employees, entrepreneurs, associations, citizens, users, to accurately predict the trends as well as identifying obstacles to the smart transformation of these small urban centers.

It is also interesting for tourism professionals, technology owners, and city administrators to begin with this research to activate a process of continuous improvement of citizen relations, business support technologies, and the implementation of other solutions and integrated services [134] that could benefit them, as well as more attention to sustainability issues, fully grasped in the three aspects (People, Planet, Profit) and improving citizen and community engagement [26,135].

**Author Contributions:** Conceptualization, G.B.; methodology, G.B.; software, G.B.; validation, A.M. and L.C.; formal analysis, G.B.; investigation, G.B.; resources, G.B.; data curation, G.B.; writing—original draft preparation, G.B.; writing—review and editing, G.B.; visualization, G.B.; supervision, A.M. and L.C. All authors have read and agreed to the published version of the manuscript.

**Funding:** This research received no external funding.

**Institutional Review Board Statement:** Not applicable.

**Informed Consent Statement:** Not applicable.

**Data Availability Statement:** Not applicable.

**Conflicts of Interest:** The authors declare no conflict of interest.

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
