# Peer review of "Small-Town Citizens’ Technology Acceptance of Smart and Sustainable City Development"

_sustainability, doi:10.3390/su15010325_

Round 1

Reviewer 1 Report

Dear Authors, 

I am pleased to review your piece of research "The acceptance of tourism technology in small towns 2 developing smart and sustainable cities". It looks like a truly promising paper, potentially interesting to a wide range of readers, thanks to the clarity, theoretical, and practical implications. 

However, there is room for significant improvement. Please let me share my suggestions below:

1.  (72-73) this sentence is too general and should be deleted or explained. What is social life in the framework of a smart city? Technology can be seen as a tool, but the culture seems to be a real foundation for a smart city, is not it? 

Are you referring to social constructivism? If "yes", could you please mention it and reflect it in the literature? If "no", please provide a clear and engaging explanation, introducing this phenomenon in detail. Now it is vague. 

2. (65-68) Could you please define "smart citizenship"? It doesn't sound serious since this construct is not properly presented, sorry. 

3. (44-50)This paragraph should be rephrased. I would invite you to pay an attention to a new book about smart cities by Buhalis et al. (2022), notably, to chapter 1 by Glebova & Lewicky "Smart cities Digital Transformation" 

Here is a link:

https://books.google.fr/books?hl=en&lr=&id=TaWaEAAAQBAJ&oi=fnd&pg=PA1&ots=MBzPdr4usQ&sig=B5Uqj0ZFsfwhNXTUnXsrDXB0fNc&redir_esc=y#v=onepage&q&f=false 

4. Sorry if I am wrong on this point, but I misunderstand subsection 2.2.: how exactly do you shape and outline the social factors, involving them in a theoretical framework? Could you please articulate it? 

5. I do clearly understand and respect your decision, however, the discussion section could reasonably take place in this paper, synthesizing theory and results and disclosing your findings in detail. 

6. Among limitations, I would mention the fact that all these factors differ a lot depending on the demographical factors. 

Author Response

Dear reviewer,

it is a pleasure to have received such valuable feedback to improve our paper. Therefore, I would like to respond to your review point by point.

1. Yes, we were referring to a constructivist approach. Then we will clarify this better by expanding the link to the literature.

2. You are right, we should not assume anything. We will broaden the concept and better explain what "smart citizens" means.

3. Thank you very much for the suggestion

4. Section 2.2 is about "social factors" influencing technology acceptance and refers to the evolution of the TAM model initially theorized by Davis (1989). This evolution considering social factors later led to the USTAM variant.

5. and 6. The discussion presented in conclusion section and limitations we will expand them according to your valuable advice.

Reviewer 2 Report

1) for abbreviation, please mention the full name the first time it is mentioned, such as USTAM.

2) why not considering to add a separate discussion of the literature review on social factors of tourists influencing technology acceptance? besides, is there any literature that discusses the socio-demographic impacts on technology acceptance?

3) the figure 1 USTAM model could be better explained, especially the subgroup of SCT and TAM variables. 

4) for all factors included in the model, could you please explain them somewhere? maybe a table could help. The same goes to the BI 1-3. How did you choose these questions? maybe supported with LR

5) better discuss more why F2 have negative impact

6) any consideration of including socio-demographic variables in the model?

Author Response

Dear reviewer,

it is a pleasure to have received such valuable feedback to improve our paper. Therefore, I would like to respond to your review point by point.

1. Okay, you are right. We will specify what USTAM means.

2. and 6. We thank you for the suggestion. We will try to consider and expand the LR by considering social factors.

3. The variables were specified when the hypotheses were stated. We may be more careful to make them clearer, though.

4. The questions chosen were extracted from the work cited in the paper, which used the USTAM model. But so we will explain it better.

5. Okay. You are right, we will better discuss why factor 2 has a negative impact on B.I. 

Round 2

Reviewer 1 Report

Dear authors,

Thank you for addressing my recommendations, I am happy to see that the paper has been significantly improved. 

However, I have noticed several points for further improvement. Could you please follow my recommendations below:

1. (116) "People, ergo community" sorry, the sentence is unclear. Please reshape it.

2. (160) "employed in several scientific studies" sounds unconvincing. It may be changed to "employed in plenty of academic research in different fields"

3. Te reference #10 is not full, since this is a chapter and it should be: 

Glebova, E., Lewicki, W. Smart cities’ digital transformation, in Buhalis, D., Taheri, B., & Rahimi, R. (Eds.). Smart Cities and Tourism: Co-creating experiences, challenges and opportunities: 548 Co-creating experiences, challenges and opportunities, 2022.   4. Future research directions are missing. It would be interesting and useful to know the authors' opinions on perspective research angles and ways to address the current study's limitations by proposing future research directions.    5. Furthermore, the research implications would be a great and logical final of abstract. Could you please add a sentence to the abstract? 

Author Response

Dear Editor, Thank you for these extra helpful recommendations for improving our manuscript. We have accepted and discussed your suggestions, and we have rectified any remaining issues in order to be published in this journal. Thank you very much.